# Early Detection of the Fungal Banana Black Sigatoka Pathogen *Pseudocercospora fijiensis* by an SPR Immunosensor Method

**DOI:** 10.3390/s19030465

**Published:** 2019-01-23

**Authors:** Donato Luna-Moreno, Araceli Sánchez-Álvarez, Ignacio Islas-Flores, Blondy Canto-Canche, Mildred Carrillo-Pech, Juan Francisco Villarreal-Chiu, Melissa Rodríguez-Delgado

**Affiliations:** 1Centro de Investigaciones en Óptica AC, Div. de Fotónica, Loma del Bosque 115, Col. Lomas del Campestre, León, Gto, C.P. 37150, Mexico; dluna@cio.mx; 2Universidad Tecnológica de León, Electromecánica Industrial, Blvd. Universidad Tecnológica #225, Col. San Carlos, León, Gto, C.P. 37670, Mexico; asalvarez@utleon.edu.mx; 3Unidad de Bioquímica y Biología Molecular de Plantas, Centro de Investigación Científica de Yucatán, A.C., Calle 43 No. 130 x 32 y 34, Colonia Chuburná de Hidalgo, Mérida C.P. 97205, Yucatán, Mexico; islasign@cicy.mx (I.I.-F.); mild@cicy.mx (M.C.-P.); 4Unidad de Biotecnología, Centro de Investigación Científica de Yucatán, A.C., Calle 43 No. 130 x 32 y 34, colonia Chuburná de Hidalgo, Mérida C.P. 97205, Yucatán, Mexico; cantocanche@cicy.mx; 5Universidad Autónoma de Nuevo León, Facultad de Ciencias Químicas, Laboratorio de Biotecnología. Av. Universidad S/N Ciudad Universitaria, San Nicolás de los Garza C.P. 66455, Nuevo León, Mexico; juan.villarrealch@uanl.edu.mx; 6Centro de Investigación en Biotecnología y Nanotecnología (CIByN), Facultad de Ciencias Químicas, Universidad Autónoma de Nuevo León. Parque de Investigación e Innovación Tecnológica, Km. 10 autopista al Aeropuerto Internacional Mariano Escobedo, Apodaca 66629, Nuevo León, Mexico

**Keywords:** surface plasmon resonance, black Sigatoka, *Pseudocercospora fijiensis*, plant pathogen, immunosensor

## Abstract

Black Sigatoka is a disease that occurs in banana plantations worldwide. This disease is caused by the hemibiotrophic fungus *Pseudocercospora fijiensis*, whose infection results in a significant reduction in both product quality and yield. Therefore, detection and identification in the early stages of this pathogen in plants could help minimize losses, as well as prevent the spread of the disease to neighboring cultures. To achieve this, a highly sensitive SPR immunosensor was developed to detect *P. fijiensis* in real samples of leaf extracts in early stages of the disease. A polyclonal antibody (anti-HF1), produced against HF1 (cell wall protein of *P. fijiensis*) was covalently immobilized on a gold-coated chip via a mixed self-assembled monolayer (SAM) of alkanethiols using the EDC/NHS method. The analytical parameters of the biosensor were established, obtaining a limit of detection of 11.7 µg mL^−1^, a sensitivity of 0.0021 units of reflectance per ng mL^−1^ and a linear response range for the antigen from 39.1 to 122 µg mL^−1^. No matrix effects were observed during the measurements of real leaf banana extracts by the immunosensor. To the best of our knowledge, this is the first research into the development of an SPR biosensor for the detection of *P. fijiensis*, which demonstrates its potential as an alternative analytical tool for in-field monitoring of black Sigatoka disease.

## 1. Introduction

The potential loss in food production is an important issue for the international community since it significantly affects the welfare of a continually increasing global population. In this sense, it is known that the low productivity of crops can be caused by several factors. However, plant diseases caused by pathogens play a significant role in production losses worldwide [1]. Principal pathogenic agents that infect plants comprise fungi, bacteria, and viruses. However, fungi receive particular interest as they have been related up to 80% of plant diseases [2].

One of the notorious plant pathologies caused by fungi is called black Sigatoka (BS) or leaf streak disease. This is the primary disease that impacts the cultivation of bananas worldwide. The causative agent called *Pseudocercospora fijiensis* has a long biotrophic phase, which prevents the observation of visible symptoms of the disease for weeks or even months, depending on the climatic conditions [3]. Generally, by the time the fungus changes to the necrotrophic phase (visible damage), the disease has spread through the crop, leading to the loss of 85 to 100% of the harvest [4]. This highly aggressive disease causes leaf necrosis, which reduces the photosynthetic area and induces premature maturation of fruits, making them unsuitable for their commercialization [4]. The Food and Agriculture Organization (FAO) estimates that between 20 and 40 percent of global crop production is annually reduced due to the damage caused by plant pests, which costs to the global economy around US$220 billion [5].

To reduce the harmful effects of plant pathogens, such as *P. fijiensis*, millions of kilograms of pesticides are used each year. This practice, while beneficial for crop yield, increases production costs and environmental pollution [6]. Therefore, the creation of a comprehensive management plan for pests and diseases is essential, particularly the development of novel analytical methods that allow the early detection of pathogens, preventing the rapid spread of diseases in crops [7]. Nowadays, the detection methods includes enzyme-linked immunosorbent assays (ELISA) and direct tissue blot immunoassays (DTBIA); molecular techniques such as polymerase chain reaction (PCR), real-time PCR (RT-PCR) and dot blot hybridization [1]; and spectroscopic and imaging techniques such as fluorescence spectroscopy, visible and infrared spectroscopy [8]. Nevertheless, the collection of the sample, time-consuming analysis and complex instruments unsuitable for field work, remain as the significant drawbacks that limit the expansion of these methods [1]. Consequently, there is a keen interest for developing fast, sensitive, *in-situ* and selective analytical devices such as biosensing systems for early detection of plant pathogens.

Some biosensors have already been reported for the detection of plant pathogens, including the *Cucumber mosaic* virus [9], *Pantoea stewartii* [10], *Plum pox virus* [11], *Prunus necrotic ringspot virus* [12], *Citrus tristeza virus* [13] and *Potato virus x* [14]. In particular, SPR biosensors have been successfully employed for detection of *Cowpea mosaic virus, Tobacco mosaic virus, Lettuce mosaic virus, Fusarium culmorum, Phytophthora infestans* and *Puccinia striiformis* [15]. However, to the best of our knowledge, no biosensor for the detection of *P. fijiensis* has been reported.

In the present work, a home-made SPR immunosensor was developed for the detection *of P. fijiensis*, the causative pathogen of plant disease black Sigatoka. For this, a glycosylphosphatidylinositol (GPI) protein (related to fungal adhesion and virulence) was extracted from the cell wall *of P. fijiensis* through a hydrofluoric acid-pyridine solution (HF-pyridine). SPR chips were functionalized with carboxylic groups via self-assembled monolayers of alkanethiols, followed by the direct immobilization of antibodies against the HF1 protein (GPI protein from *P. fijiensis*) onto gold-coated SPR chips. The interaction between samples spiked with several concentrations of HF1 and antibodies immobilized onto the chips was followed in real time by SPR. The analytical parameters of the biosensor were established, obtaining a sensitivity of 2.1 ng mL^−1^ with no matrix effects observed during its performance using real leaf banana extracts. This research represents the first approach to the fabrication of robust and reusable SPR platform for routine monitoring and early detection of plant disease black Sigatoka in banana plantations. 

## 2. Materials and Methods

### 2.1. Chemicals and Immunoreagents

The organic solvents for cleaning the gold-coated chips (acetone and ethanol) and the reagents used for gold functionalization (16-mercaptohexadecanoic acid (MHDA) and 11-mercaptoundecanol (MUD)) and immobilization (ethanolamine hydrochloride, N-hydroxysuccinimide (NHS), 1-ethyl-3-(3-dimethylaminopropyl)carbodiimide hydrochloride (EDC)) and cell wall extraction, phenyl methyl sulfonyl fluoride (PMSF), Ethylene glycol-bis(2-aminoethylether)-N,N,N′,N′-tetraacetic acid (EGTA) and sodium dodecyl sulfate (SDS), were purchased from Sigma-Aldrich (St. Louis, MO, USA). 

*Buffers and solutions*. The antigen and antibodies were dissolved in phosphate buffer saline (PBS): 2.7 mM potassium chloride, 10 mM phosphate buffer and 137 mM sodium chloride, pH 7.5. The immobilization reagents (EDC/NHS) were dissolved in 2-(N-morpholino) ethanesulfonic acid (MES) buffer: 100 mM, 500mM NaCl, pH 5.5. For the obtention of the antigen an extraction buffer (PBS pH 7.4, supplemented with 1% SDS and 1 mM PMSF) and a Tris-HCl buffer, pH 7.4 (supplemented with 50 mM NaCl, 10% glycerol, 1 mM EDTA, 1 mM PMSF, 1 mM EGTA, 5 mM β-mercaptoethanol, 1 µg/mL leupeptine, 1 µg/mL aprotinine, 1 tablet/L protease inhibitor cocktail (Roche^TM^, Mannheim, Germany, 0.03 g polyvinylpolypyrrolidone/mL), were employed. The salts employed in buffer preparation were purchased from Sigma-Aldrich.

*Fungal material*. *Pseudocercospora fijiensis* fungal strain C1233 (registration number IMI 392976, International Mycological Institute, CABI Bioscience Centre, Egham, UK) was isolated from a *Musa acuminata* plantation located in the southern state of Yucatan, Mexico (20°25′36″N; 89°45′20.3″W). Briefly, the fungus was grown in potato dextrose broth supplemented with 200 mL L^−1^ V8 vegetable juice (Herdez^®^) and incubated at room temperature at 100 rpm. After 11 days of incubations, the mycelium was harvested by filtration through two layers of cheesecloth, washed twice with sterile water and powdered with liquid N_2_ on a sterile mortar and pestle.

*Immunoreagents*. The HF1 protein (GPI protein isolated from *P. fijiensis* cell wall) and polyclonal antibody (anti-HF1) were prepared by the Unit of Biochemistry and Molecular Biology of Plants (CICY, Yucatan). The methodology for the obtention of the GPI proteins from the cell wall of *P. fijiensis* was adapted from the method described by Maddi et al. [16] as follows: the powder obtained from the fungus mycelium was split in portions of 2 g, then each portion was homogenized with 1 mL of extraction buffer. The homogenate was centrifuged at 4000 rpm for 5 min, followed by a washing step of the pellet (cell wall) with extraction buffer. Then, the pellet was suspended in extraction buffer, heated by 15 min at 95 °C, cooled in ice by 5 min and finally centrifuged at 10,000 rpm during 5 min. The resulting supernatant was discarded and the cell wall (pellet) was washed thrice with extraction buffer, twice with 1 mL of cold sterile distilled water containing 1 mM PMSF and finally lyophilized. 

The lyophilized cell wall (0.5 g) was treated with 1 mL of 30 mM NaOH at 4 °C for 4 h (under gently agitation) and stopped by adding an equal volume of glacial acetic acid and incubation of 2 h. The mixture was centrifuged at 16,000 rpm at 4° C during 5 min, supernatant was discarded, and cell wall (pellet) was washed with distilled deionized water. Then, the pellet was treated with 1 mL hydrogen fluoride-pyridine (HF-pyridine 70:30 v/v) and incubating for 2 h to release the cell wall bound GPI-proteins. The reaction was stopped by adding equal volumes of sterile water and incubating by 90 min in ice. Then, the mixture was centrifugated at 14,000 rpm during 5 min, the GPI-released cell wall proteins were recovered in the supernatant. The supernatant was dialyzed overnight against 4 L of sterile deionized water, using Slide-A-Lyzer filter (Thermo Scientific, Rockford, IL, USA). After dialysis, proteins were pooled, added 0.015% sodium deoxycholate (DOC) and precipitated overnight with 10% of trichloroacetic acid (TCA) at 4 °C. Precipitated proteins were collected by centrifugation at 16,000 rpm, washed twice with cold acetone (1 mL by tube). Then, the pellets were suspended in buffer PBS and maintained at −20 °C. A stock solution of HF1 at a concentration of 150 µg mL^−1^ was prepared from a lyophilized powder of the antigen dissolved in PBS. 

The proteins´ profile was obtained by sodium dodecyl sulfate-polyacrylamide gel electrophoresis (SDS-PAGE) and further N-terminal sequencing of individual bands of the gel. The resulting 22 kDa protein, identified as HF1, was excised from the gels and subjected to electroelution according with Brito-Argáez et al. [17], to recover the protein and employed it for the antiserum. The antiserum production is described below: 200 μg of HF1 protein was emulsified in complete Freund’s adjuvant, which was used to immunize subcutaneously New Zealand white rabbits (6 months´ old, female). Two subsequent booster injections of the immunogen in incomplete Freund’s adjuvant were made at 2-week intervals. The blood was collected from the ear vein one month after the last immunization. The immune serum was recovered from rabbit and IgGs were separated from the rest of blood cellular proteins by using protein A-Sepharose column chromatography, according with the instructions of the manufacturer (BioVision, Mountain View, CA, USA). The polyclonal antibodies obtained were denominated as anti-HF1.

*Preparation of leaf extracts*. The leaf extracts were obtained from leaves of ‘Grand Nain’ (*Musa acuminate*) plants grown in a greenhouse under conditions previously described [4]. The leaves were washed with running water and treated with sodium hypochlorite (0.6 %; v/v), then cut into small pieces, mixed with sterilized distilled water and further blended in 50 mM Tris-HCl pH 7.4 supplemented. The homogenates were placed on ice by 15 min, then centrifuged at 14,000 rpm and the supernatant was lyophilized.

### 2.2. Chromium and Gold Thin Film Deposition

Gold (Au) and chromium (Cr) pellets with a purity of 99.99% were used to evaporate over glass substrates. The first layer of chromium was evaporated by electron gun evaporation technique with a thickness of 3 nm; in the same vacuum chamber, gold was evaporated by thermal evaporation technique at a rate of 5 Å/s in an atmosphere of 8 × 10^−6^ mbar with a thickness of 50 nm, which was corroborated using a quartz crystal microbalance thickness monitor (XTC/2 Depositions Controllers Leybold Inficon, San Jose, CA, USA). To ensure homogeneous thicknesses of the Cr/Au thin film deposited on thin glass substrates, they were placed perpendicular to the longest distance (approximately 36 cm) from the evaporation source in the vacuum chamber and placed the closest to the thickness monitor. The Cr/Au chips were adhered to the prism (BK7) using oil matching index (*n* = 1.51).

### 2.3. SPR Setup

The SPR system employed in this study was modified from the instrument described previously by Sánchez-Alvarez et al. [18]. In short, the optomechanical setup was a homemade platform based on the Kretschmann configuration, consisting of two stacked and motorized rotation plates of 0.00025° of angular resolution, configurated to move synchronized according a θ–2θ system (one plate moves at twice the speed and reaches twice the degree rotation than the other) by a stepper motor. A hemicylindrical shaped glass prism BK7 was mounted on the base of the superior plate, meanwhile the photodetector (Si photodiode model S1226-8Bk with a signal amplifier circuit Hamamatsu, Bridgewater, NJ, USA) was assembled on the lower plate; allowing the capture of the reflected light, launched by a 632.8 nm p-polarized He-Ne laser (mod. 1101P, Uniphase, St. Charles, IL, USA) and that passes through the prism (Figure 1a). The intensity of the reflected light captured by the photodetector is later transformed into a voltage (range from 0 to 12 V) and collected using a DAQ device (USB 6003, 16 bits, National Instruments Mexico, Ciudad Juárez, México) with a resolution of 1.8 ×10^−4^ V·bit^−1^. The fluidic system consisted of a Teflon cell (depth of 1 mm and area of 16 mm^2^) with an inlet and outlet tube placed at 45 degrees. The entrance tube of the cell possesses four syringe tips (0.9 mm diameter) that allows the simultaneous injection of different solutions (Figure 1b,c). However, in this study only one syringe was employed, meanwhile the rest were blocked. The design of the Teflon cell allowed a laminar flow through its inner channel which possesses a diameter of 1.4 mm. The gold coated surface of the chip was pressed against the flow cell to allow the solutions came in contact with the gold. Meanwhile, the glass surface of the chip was optically matched to the prism (with its flat face downward) by using immersion oil (Figure 1d). The solutions were continuously delivered over the sensor surface by using a syringe pump (Legato 100) at a flow rate of 30 μL min^−1^. The communication between the SPR setup and the PC was achieved via an RS-422 serial link and controlled by a homemade graphical user interface programmed in LabView (National Instruments Mexico, Ciudad Juárez, México) to allow the data acquisition. The SPR setup possess an angular sensitivity of 172°/RIU, (using a BK7 prism coupled to a Cr/Au thin film) and a signal-to-noise ratio (SNR) of 7.42 db (decibels).

### 2.4. SPR Chip Biofunctionalization

First, a cleaning step was performed on the Au/Cr chip surfaces before the biofunctionalization. Chips were consecutively immersed in acetone and ethanol for intervals of 30 s, for each solvent, and then dried with air. Once the chips were clean, a mixed self-assembled monolayer (SAM) was formed by incubating the gold substrates overnight at room temperature with a solution of alkanethiols MHDA:MUD (total concentration of 250 µM in ethanol). 

Then, the anti-HF1 was immobilized onto the gold surface using the EDC/NHS method [19]. Briefly, the immobilization of the antibody was accomplished by attaching the free amine groups of the anti-HF1 to the carboxylic terminal groups of the alkanethiol monolayer through an amide bond. The biofunctionalization was performed in-flow by delivering a solution of EDC/NHS (EDC 0.2 M/NHS 0.05 M) in MES buffer (100 mM, 500 mM NaCl, pH 5.0), followed by the injection of a solution of anti-HF1 (50 µg mL^−1^). Finally, a solution of ethanolamine (1 M, pH 8.5) was flowed over the sensor surface to allow the deactivation of the remaining unreacted carbodiimide esters. A conceptual overview of the immobilization procedure is observed in Figure 2.

### 2.5. Immunoassay Procedure-HF1 detection

All measurements were based on a direct immunoassay where the immobilized antibodies anti-HF1 binds to the antigen HF1 in the sample. PBS buffer and leaf extracts were spiked with different amounts of antigen, ranging from 0 to 122 µg mL^−1^, by sequential dilution from a stock solution of HF1 at a concentration of 150 µg mL^−1^. In the study, PBS was set as running buffer. Samples containing HF1 (300 µL volume) were flowed at 30 µL min^−1^ over the sensor surface, and the binding events were subsequently monitored. Then, the sensor surface was washed with the injection of PBS buffer during 8 min to remove weakly bound of HF1 to the biofunctionalized chip. The obtained signals (antibody-antigen binding) were directly proportional to the concentration of the analyte in the samples, observed as a change in the intensity of the light measured by the photodetector (see Figure 2). The obtained average SPR signal (*n* = 3) were plotted as a function of HF1 concentrations to generate a calibration curve. A regeneration solution of NaOH 20 mM (injected during 20 s) was employed to remove the bound of the protein, obtaining a reusability of the surface for 10 cycles. The calibration curve was fitted throughout the entire measured concentration range of antigen with linear regression analysis and employed to establish the analytical parameters of the biosensor. The limit of detection and the limit of quantitation were calculated as 3 and 10 times the standard deviation of the baseline, respectively (LOD, = 3 * SD; LOQ, = 10 * SD) and the sensitivity as the slope of the curve. The dynamic range defines the span of the maximum and minimum concentration (i.e., the limit of quantification) of analyte measured by the biosensor, with an extent to a range where linear relationship between the concentration measured and sensor signal is maintained [20].

## 3. Results and Discussion

For this study, an SPR-based immunoassay employing polyclonal antibodies was proposed for the label-free detection of HF1, a protein from *P. fijiensis*, the causative pathogen of black Sigatoka disease. In this scheme, the antibodies were immobilized onto a gold-coated chip, with a specific concentration of the target analyte flowed over the biofunctionalized sensor surface. The binding event due to the interaction of the analyte and the anti-HF1 layer generated a signal directly proportional to the HF1 concentration in the sample.

### 3.1. SPR Chip Biofunctionalization

Long chain alkanethiols (mercaptohexadecanoic acid, MHDA; and mercaptoundecanol, MUOH) were employed to form a self-assembled monolayer (SAM) onto the gold layer substrate for its further use as a platform for the immobilization of HF1 polyclonal antibodies. According to Cheng et al. [21] the density of a mixed self-assembled monolayer (SAM) of alkanethiols (nitrilotriacetic acid thiol and oligo(ethylene glycol thiols), measured by XPS, ranged from 0.9 to 1.3 molecule/nm^2^. In our work we solely evaluated the formation of the SAM with terminal COOH through an indirect methodology published by [19,22], where the efficiency in the process of immobilization (which occurs through the activation of COOH moieties from the SAM) is measured in real time at fixed angle). In this case, the increment in the SPR reflectance corresponds to the increase of thickness in the film due to the density of immobilized molecules. The density of molecules immobilized on a surface for later use as a bioreceptor is an essential factor to consider for efficient detection; since a low density could lead to a poor recognition, meanwhile excessive amounts can result in steric hindrance, causing a deficient detection as well [22]. Thus, several antibody concentrations (anti-HF1 = 10, 30, 50, 60 and 70 µg mL^−1^) immobilized onto two molar ratios of a mixed SAM (MHDA: MUD = 1:20 and 1:50, total alkanethiol concentration 250 µM) were tested, obtaining maximum responses by using a ratio 1:20 of MHDA: MUD and 50 µg mL^−1^ of anti-HF1 (see Figure 3).

The entire immobilization process was performed in-flow (using water as the running solution) at a fixed angle of 68.5, which was obtained from the angular sweep SPR curve measured in water (Figure 4b). This angle is the point with the highest SPR sensitivity to the changes in reflectivity due to binding events over the surface, (angle located in the middle of the maximum steepest slope of the SPR curve in water). Thus, the immobilization was real-time monitored at θ = 68.5 by the SPR setup, obtaining a representative sensogram as observed in Figure 4a.

First, surface activation of the SAM carboxylic acid groups occurs by injection of EDC/NHS, followed by immobilization of anti-HF1 by a carbodiimide-mediated bond to exposed amine groups, establishing an amide bond. After immobilization, the surface of the sensor was blocked with ethanolamine to avoid nonspecific binding during subsequent measurements, followed by a washing step with water to remove weakly bound molecules. The increase in the level of the SPR signal after the injection of each reagent (compared to the baseline of the running solution) (see Figure 4) allowed to infer the successful linkage of the molecules that will served as the anchorage for the anti-HF1 (bioreceptor).

The Figure 4b shows the SPR curve in water of a gold-coated chip before and after immobilization, showing a shift of 1.3 degrees (from 71.4° to 72.7°) in the resonance angle due to the quantities of bound antibodies. The shift is attributed to the increase of surface mass density due to the linkage of the antibodies on the chip. According several studies, the immobilization densities have been calculated using a conversion factor of 1 ng/mm^2^ of biomolecule/protein, which corresponds to a change of 1000 RU (refractive units) or a shift of 0.1° in the response angle [23,24,25]. In this context, the shift of angle obtained after the immobilization was 1.3° which represents a density of 13 ng/mm^2^. These results are in a good agreement with the ones observed in the sensogram of the process of immobilization measured in real-time, suggesting a successful biofunctionalization. 

### 3.2. Immunoassay Procedure-HF1 Detection

A direct binding immunoassay was performed under the aforementioned assay conditions. Different concentrations of HF1 ranging between 0 and 122 µg mL^−1^ were flowed over the anti-HF1 coated surface, followed by a washing step with PBS buffer once each concentration reached the plateau, saturation level in the signal response. The measurements were performed in triplicate and the average was fitted to a centered fifth order polynomial equation (y = β_0_ + β_1_*x* + β_2_*x*^2^…+ β_5_*x*^5^), where *X* is the value from the subtraction of the mean X from all X values before fitting and β are the values of Poisson model [26]. This fitting method is commonly employed to smooth and estimate the trajectory shape of curves from a group of data, in this case, the sensograms (see Figure 5). The analysis was based on least squares, obtaining in all the fittings a degree of correlation of R^2^ higher than 0.96. Then, the SPR signal value at the plateau (saturation by specific binding) obtained from the fitted trajectory curves were plotted as a function of HF1 concentrations to generate a calibration curve (see Figure 5). The limit of detection obtained was 11.7 µg mL^−^^1^ which is comparable to that obtained by Torrance et al. [27], where used an antibody-based SPR for the detection of *Cowpea mosaic virus*, establishing a limit of detection of 12.5 µg mL^−^^1^ as well as the LOD of 10 µg mL^−1^ obtained by Otero et al. [28] who employed a monoclonal antibody-based on a triple antibody system-ELISA for the detection of *Mycosphaerella fijiensis*. The primary analytical parameters obtained in this study are summarized in Table 1. Currently, there is a lack of analytical tools that address the detection of fungus and bacteria as plant pathogens, since SPR-biosensors have been mainly limited to detection of plant viruses such as *Cymbidium mosaic virus, Odontoglossum ringspot virus* [29], *Cowpea mosaic virus* [27], *Tobacco mosaic virus, Lettuce mosaic virus* [30]. In particular, this research represents the first approach to the fabrication of robust SPR platform for routine monitoring and early detection of *P. fijiensis*, the causative agent of plant disease black Sigatoka in banana plantations. 

### 3.3. Evaluation of SPR Performance with Real Banana Leaves Extract—Study of Matrix Effect

Since black Sigatoka leaf disease has a biotrophic phase, could pass weeks or even months before visible symptoms in the leaves of the banana plants. For this reason, it is necessary to assess and monitor the presence of the concentrations of HF1 in the extracts from the leaves, as indicative of the possible infection by *P. fijiensis*. Thus, the evaluation of the matrix effects in the SPR based immunoassay is a critical issue to determine its influence on the performance of the method during the analysis of real samples. The presence of certain components in the matrix of a real sample could lead to false positives as a consequence or interfere in the antigen-antibody recognition [22].

Real banana leaves were collected and processed as aforementioned to obtain aqueous extracts, which were directly injected in the SPR setup to observe the sensor response (see Figure 6). This method is widely employed, since the mechanical disintegration of the leaves allows the release of the fungus *M. fijiensis*, exposing out the proteins in the cell wall for its further detection through the antibodies [4,28,31]. 

For comparison, a sample of the aqueous extract was spiked with a concentration of 90 µg mL^−^^1^ of HF1, considering that the amount of analyte is required to fit the dynamic range obtained from the calibration curve in analytical parameters (see Table 1). 

As shown in Figure 6 no matrix effect is observed when a 100% of the extract was added. Thus, samples from leaves extracts do not need to be diluted or pretreated (centrifugation, purification or extraction steps) to be measured by the SPR immunoassay. In contrast with other reported methods, such as ELISA and PCR. On the other hand, the result obtained for the spiked sample was a concentration of 87.7 µg mL^−^^1^, showing a percentage recovery of 98.5%. The proposed method showed the potential of the device to achieve the early detection of black Sigatoka disease in banana plantations to avoid its damage. The analysis time of the device was 40 min and not sample pretreatment is required, which facilitates the measurements.

## 4. Conclusions

The home-made SPR immunosensor fabricated in this work was successfully tested on banana leaves extracts. It exhibited a LOD for the HF1 antigen of 11.7 µg mL^−1^ and a sensitivity of 0.0021 units of reflectance per ng mL^−1^. These parameters along with the absence of matrix effects observed during its performance using real extracts demonstrate its potential as an attractive analytical tool for in-field detection of the black Sigatoka disease. The application of this biosensor would improve the disease management by monitoring the crop health in real-time, generating databases that would furtherly enhance the production yield. It is important to emphasize that, despite this research establishes the first SPR biosensor that addresses the challenges associated with the detection of *P. fijiensis*, future research needs to look at the assessment of other factors that might affects the detection protocol, such as fungal protein concentration in the leaves at each phase of the disease, plant ageing, climatic conditions and possible interferences with pathogen agents such as *M. musicola*, *M. musae* and *M. minima*.

## Figures and Tables

**Figure 1 sensors-19-00465-f001:**
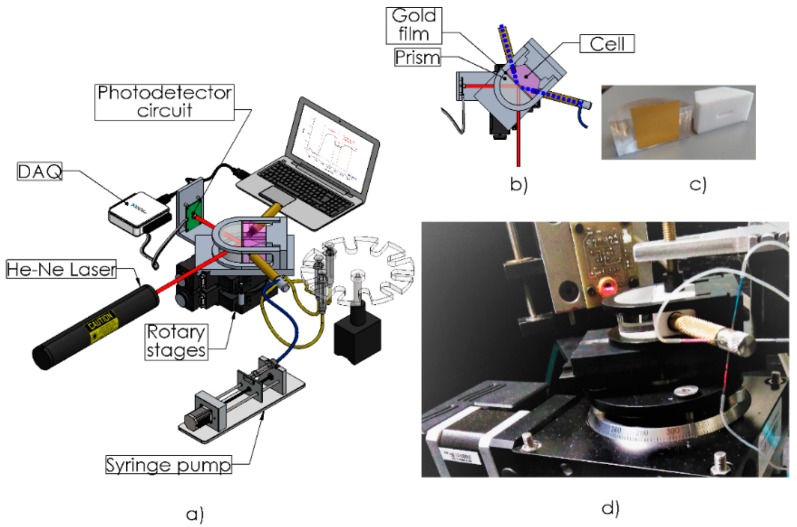
(**a**) SPR optomechanical setup; (**b**) Scheme of the Flow cell mount and (**c**) Teflon cell with gold film adhered to prism; (**d**) connections to the injection system.

**Figure 2 sensors-19-00465-f002:**
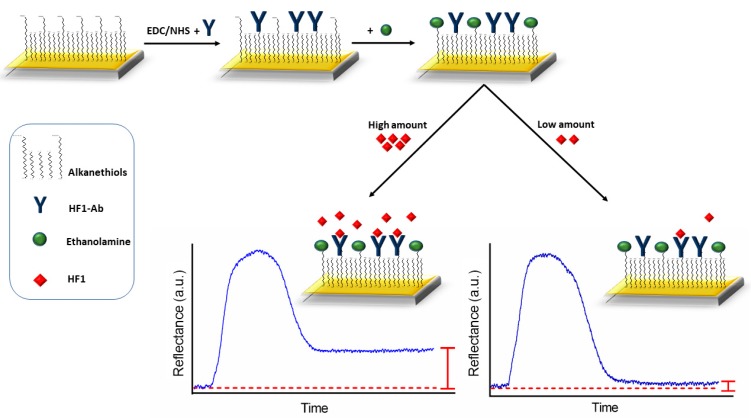
Schematic of the SPR immunoassay for detecting HF1 and illustrative representation of sensograms.

**Figure 3 sensors-19-00465-f003:**
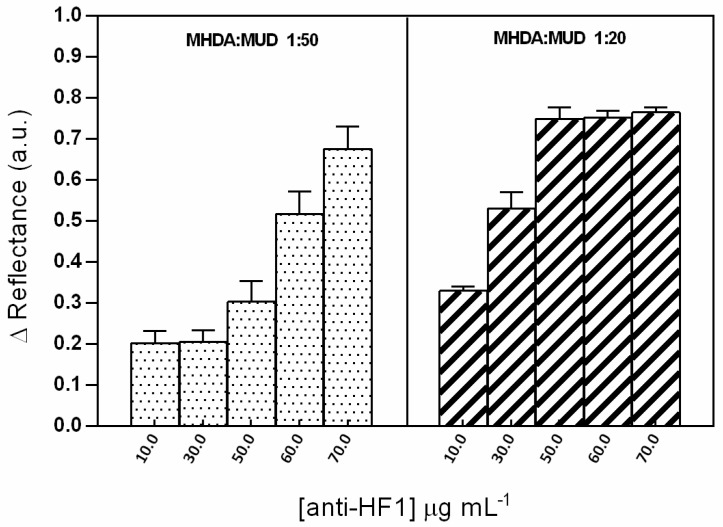
SPR response as a change in the reflectance during the immobilization process employing several antibody concentrations (10, 30, 50, 60 and 70 µg mL^−1^) and two molar ratios of a MHDA: MUD (1:20 and 1:50). Each point represents the mean ±SD of three measurements.

**Figure 4 sensors-19-00465-f004:**
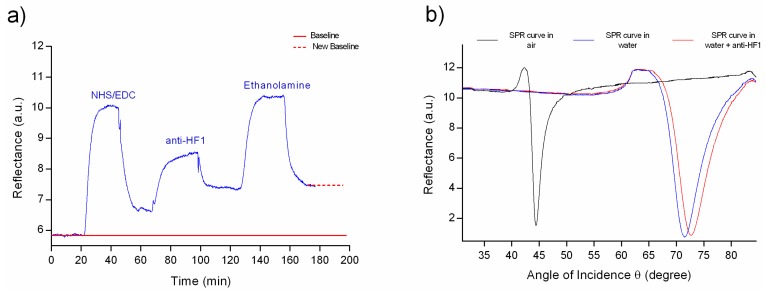
(**a**) Real-time SPR sensorgram of anti-HF1 immobilization. Activation by EDC/NHS; then immobilization of the polyclonal antibodies and finally blocking of the sensor surface by ethanolamine and (**b**) Angular reflectance spectra measured from a sensor chip before (blue) and after the immobilization (red) of anti-HF1.

**Figure 5 sensors-19-00465-f005:**
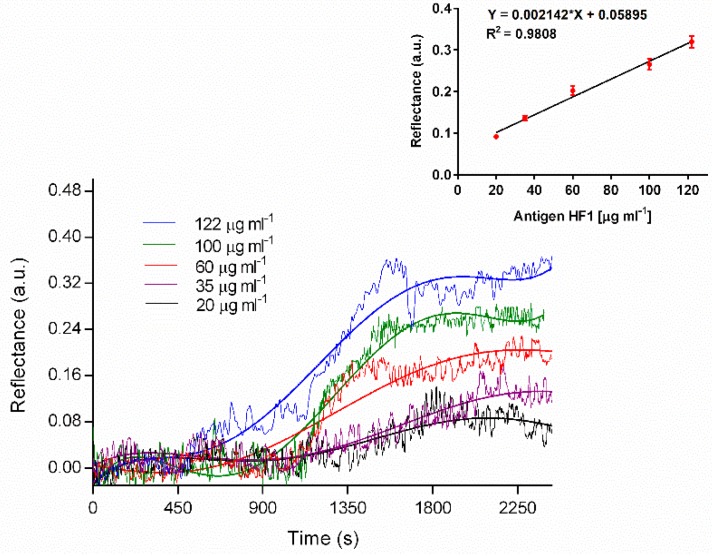
Real-time SPR sensograms for HF1 detection at different concentrations and calibration curves in PBS. Each point represents the mean ± SD of three replicates.

**Figure 6 sensors-19-00465-f006:**
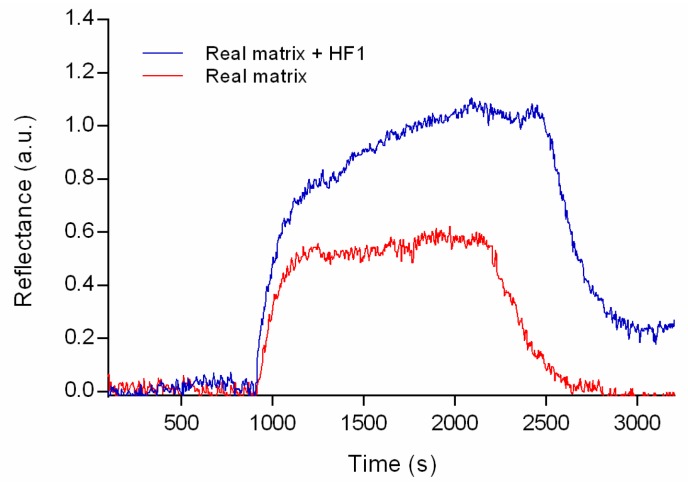
SPR sensograms evaluating the level of nonspecific signals due to matrix effects resulting from banana leaves extracts.

**Table 1 sensors-19-00465-t001:** Analytical parameters of SPR based biosensors for black Sigatoka pathogen detection.

Pathogen	Infected Crop	LOD (µg mL^−1^)	LOQ (µg mL^−1^)	Sensitivity (units of reflectance/ng mL^−1^)	Dynamic Range (µg mL^−1^)
*P. fijiensis*	Banana	11.7 ± 0.01	39.1 ± 0.01	0.0021 ± 0.0001	39.1–122

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
