# Peer review of "Early Detection of the Fungal Banana Black Sigatoka Pathogen Pseudocercospora fijiensis by an SPR Immunosensor Method"

_sensors, 2019, doi:10.3390/s19030465_

Reviewer 1 Report

Authors of the manuscript entitled “Early detection of fungal banana black Sigatoka pathogen Pseudocercospora fijiensis by an SPR immunosensor method” demonstrated very interesting work  for the detection of the Pseudocercospora fijiensis, that is very problematic pathogen especially for cultivation of bananas. To my best knowledge this is first work aiming at the detection of this pathogen.

I this this work could published after certain improvements.

First of all, I want to emphasize that the sensing methodology used by authors (SPR chips, SAM growth, EDC/NHS enabled immobilization) are very commonly used. Therefore, the novelty of this work is not about sensing principles but the extraction of pathogens, and the immunosensing properties itself. Thus, in order to be more in line with sensors audience, authors should present more details on the propertirs of their sensors:

1)      Surface quality? The growth of SAM layers may affect the properties of the sensors. Authors should present the density of the COOH groups. XPS at high angle should help.

2)      Immobilized antibodies, what is the density of the antibodies? Again XPS or ToF-SIMS

3)      What about stability and selectivity of the sensors? Authors can show the results when the probe of different pathogen or mixture of pathogens is analyzed.

Small remark about introduction. Authors may exclude the introduction of standard procedure, e.g. SPR principles. Sensors audience supposed know this information, but the reference to reasonable review papers is welcome, of course. I would like to note that the length of the paper is quite big, although authors did not show the surface analysis and sensors selectivity results. I do recommend during second revision to reduce the size of the paper.

Author Response

Reviewer #1: General Comments:

Authors of the manuscript entitled “Early detection of fungal banana black Sigatoka pathogen Pseudocercospora fijiensis by an SPR immunosensor method” demonstrated very interesting work for the detection of the Pseudocercospora fijiensis, that is very problematic pathogen especially for cultivation of bananas. To my best knowledge this is first work aiming at the detection of this pathogen.

I this this work could published after certain improvements.

First of all, I want to emphasize that the sensing methodology used by authors (SPR chips, SAM growth, EDC/NHS enabled immobilization) are very commonly used. Therefore, the novelty of this work is not about sensing principles but the extraction of pathogens, and the immunogens properties itself. Thus, in order to be more in line with sensors audience, authors should present more details on the properties of their sensors:

Specific Comments:

1)     Surface quality? The growth of SAM layers may affect the properties of the sensors. Authors should present the density of the COOH groups. XPS at high angle should help.

We understand the suggestion made by the reviewer, especially since the COOH are the base for covalent linkage of the free amino group in the bioreceptor (immobilization of antibodies). However, the creation of the Self-assembled monolayer with terminal COOH was evaluated indirectly through the efficiency in the process of immobilization, which was monitoring by the SPR system. This indirect evaluation has been performed following the methodology published by [1], [2], where the immobilization is monitored in real time at fixed angle and also as the angle shift in angular sweep (represented in Fig 4). In this study, two ratio of alkanethiols MHDA: MUD 1:20 and 1:50 were tested, obtaining maximum signal responses at 1:20. The results of the immobilization factors tested during the study are summarize in an added graph in the manuscript (Figure 3). 

2)      Immobilized antibodies, what is the density of the antibodies? Again XPS or ToF-SIMS

The reviewer is right, the density of antibodies immobilized reveals important information since a low density could lead to a poor recognition, meanwhile excessive amounts can result in steric hindrance, causing a deficient detection as well. However, the immobilization of the antibodies does not determine the correct orientation nor functionality of the antibody. Thus, the quality/ activity of the biofunctionalized surface was measured directly through the efficiency in the immunoassay, following the methodology published by [1], [2] which is widely employed in SPR biosensing studies, where the immobilization is monitored in real time at fixed angle and also as the angle shift in angular sweep (represented in Fig 4). In this study, several antibody concentrations (anti-HF1 = 10, 30, 50, 60 and 70 µg mL-1).  The results of the immobilization factors tested during the study are summarize in an added graph in the manuscript (Figure 3).  However, in several studies, the immobilization densities have been calculated using a conversion factor of 1 ng/mm2 of biomolecule/protein corresponds to a change of 1000 RU (refractive units) or a shift of 0.1° in the response angle [3]–[5]. In this context, the shift of angle obtained in this study after the immobilization was 1.3°, which represents a density of 13 ng/mm2. Please refer to lines

3)     What about stability and selectivity of the sensors? Authors can show the results when the probe of different pathogen or mixture of pathogens is analyzed.

This work was focused in the development of a detection protocol at controlled conditions in order to show the potential of the technique in the early detection (asymptomatic phase) of black sigatoka disease. Nevertheless, more experimental evidence would be required before the propose of the method as a high scale monitoring system. The authors are currently working on another study addressing other factors that affects the detection protocol, such as fungal protein concentration in the leaves at each phase of the disease, plant ageing, climatic conditions and possible interferences with pathogen agents M. musicola, M. musae and M. minima.

 Small remark about introduction. Authors may exclude the introduction of standard procedure, e.g. SPR principles. Sensors audience supposed know this information, but the reference to reasonable review papers is welcome, of course. I would like to note that the length of the paper is quite big, although authors did not show the surface analysis and sensors selectivity results. I do recommend during second revision to reduce the size of the paper.

Several sentences were eliminated to improve the manuscript. Please refer to lines 112

Reviewer 2 Report

The authors summarize the development of a SPR based immunoassay for the detection of a protein present on the P. fijiensis pathogen. They have also shown some preliminary results on the matrix effect from banana leaves extracts, which is a crop where these fungus can be found. I found however some aspects that need to be addressed before considering the acceptance of the paper.

-The description of the instrument is fine although figure 1 is not clear enough, specially 1c. It is too small and dark. The authors must improve the size and quality of the picture. Moreover, the description of the fluidic part is confusing, so the Teflon cell is not well depicted in figure 1b. it seems from the description it is a single channel, but the entrance tube has “4 syringe tips to allow simultaneous injection of several solutions”. Please clarify this as the design is confusing. Are the authors introducing several samples at the same time besides the buffer that flows continuously?

-Figure 2 shows the detection of two solutions for high and low analyte concentration. Both cases there is an initial increase in the SPR signal before stabilizing. As they describe in section 2.5,  PBS is set as running buffer and the samples are diluted in the same buffer. Where does then this initial increase of the SPR signal come from? Please clarify this. Later on, in figure 4 this initial increase in the signal is not observed.

-The analytical parameters of the fitting are not clear, especially the sensitivity value. If it is the slope of the curve, the units they provide are not correct. Please elaborate the explanation and the description of this.

-What is the noise of the system?

-The section 3.1, where they explain the biofunctionalization, is not well described. Please describe why you measure at 68.5 degrees. The figure 1b and its discussion is confusing. So these results have been obtained with two different chips? Is that correct? Or before and after biofunctionalization? Is this shift after immobilization (71.4 to 72.7 degrees) proportional with the signal obtained in figure 3.a?  It seems too big of a shift for what it is observed in figure 1a.

-The authors must show more data to support their results. For example, they detailed some immobilization factors they studied (i.e.  SAM ratios and concentration of antibody). They just state that they select the conditions as they gave “maximum responses”. They must summarize the results for the other conditions as well, to support this conclusion and show some graphs/tables too. Looking at figure 3a. they do not indicate to what conditions (SAM and antibody concentration) this sensorgram corresponds to, but assuming is for the best ones, the immobilization signal is quite small. Moreover, if we check the detection in figure 4a. The concentrations used are considerably high and the signals very low, with a lot of noise.

-The calibration curve in Figure 4 should be bigger and all the fitting parameters should be included. It also shows the detection of the protein at different concentration. The authors must clarify if each measurement has been obtained with a different chip or with the same. If it is the same chip, have they regenerated the surface? how? What conditions? And how stable is then the surface?

-The matrix effect study is somehow incomplete. They must clarify if the extraction process they follow is the one that is commonly done to isolate the fungus, otherwise is useless. They do this process and afterwards, they spike the extract with the protein. In this case, the recovery is close to 100%. But it is not clearly explained if this process is enough to extract the fungus and what is the actual recovery of the methodology. Moreover, it is not clear if with this simple process is enough to detect the protein. Is it necessary to later extract also the protein from the cell wall? or are the antibodies capable of recognizing the protein directly in the fungus? Have this been evaluated? The authors should explain better this before assessing if this preliminary evaluation is useful or not.

-The last paragraph before the conclusions they claim the potential of the method to achieve “early detection of the disease and that not sample pretreatment is needed”. This must be clarified in both terms: what are approximately the LOD requirements for early dtection? And what do they mean that no pretreatment is needed?

Author Response

Reviewer #2 General Comments:

The authors summarize the development of a SPR based immunoassay for the detection of a protein present on the P. fijiensis pathogen. They have also shown some preliminary results on the matrix effect from banana leaves extracts, which is a crop where these fungus can be found. I found however some aspects that need to be addressed before considering the acceptance of the paper.

Specific Comments:

1)     The description of the instrument is fine although figure 1 is not clear enough, specially 1c. It is too small and dark. The authors must improve the size and quality of the picture. Moreover, the description of the fluidic part is confusing, so the Teflon cell is not well depicted in figure 1b. it seems from the description it is a single channel, but the entrance tube has “4 syringe tips to allow simultaneous injection of several solutions”. Please clarify this as the design is confusing. Are the authors introducing several samples at the same time besides the buffer that flows continuously?

The quality of the figure 1 has been improved. An image of the Teflon cell has been added (1c). The description of the fluidic system has been restructured and has been reflected in the manuscript, please refer to lines 218-222

2)     Figure 2 shows the detection of two solutions for high and low analyte concentration. Both cases there is an initial increase in the SPR signal before stabilizing. As they describe in section 2.5,  PBS is set as running buffer and the samples are diluted in the same buffer. Where does then this initial increase of the SPR signal come from? Please clarify this. Later on, in figure 4 this initial increase in the signal is not observed.

SPR sensors are highly sensitive to changes of the refractive index occurring over the sensor surface, causing a decrease or increase in the intensity of the light reflected (depending on the refractive index flowed over the sensing surface). The concentration of antigen in the samples causes changes in the solution density thus differences in their refractive index, which can be observed as different increases despite the samples are diluted in the same buffer.

However, we would like to clarify that the sensograms in the figure 2 are not from the measurements performed during the study but just an illustrative representation to explain the fundament/principle of the technique employed. This has been clarified in the caption of the Figure 2, please refer to line 252

3)     The analytical parameters of the fitting are not clear, especially the sensitivity value. If it is the slope of the curve, the units they provide are not correct. Please elaborate the explanation and the description of this.

The reviewer is right. The sensitivity units have been corrected in the Table 1 and the explanation of the analytical parameters was restructured, please refer to lines 267-269.

4)     What is the noise of the system?

The biosensors produce an optical or electronic signal (analog signal) proportional to the specific interaction between the analyte and the recognition molecule present in the biosensor, which is recorded and transformed to a digital signal through an analog-digital converter (ADC). The output signal of the biosensor system is limited by noise, due to multiple causes, such as voltage variations, electronic detection circuitry, temperature variations, flow variations, turbulent motion of the medium within the measuring cell, as well as by variations in laser output power and polarization [6]. To improve this signal, we employed digital filtering, after the output signal of the biosensor system. When the signal is a DC signal, the Signal-to-Noise Ratio (SNR) Signal Amplitude can be calculated using the signal average of amplitude and Noise Amplitude can be calculated using the standard deviation of the measured signal. This SNR calculation method was implemented in Matlab resulting in 7.42 db (decibels) in the measured signal from the sensor.  please refer to lines 229-230.

5)     The section 3.1, where they explain the biofunctionalization, is not well described. Please describe why you measure at 68.5 degrees. The figure 1b and its discussion is confusing. So these results have been obtained with two different chips? Is that correct? Or before and after biofunctionalization? Is this shift after immobilization (71.4 to 72.7 degrees) proportional with the signal obtained in figure 3.a?  It seems too big of a shift for what it is observed in figure 1a.

The entire process of immobilization was performed in-flow (using water as the running solution) at a fixed angle of 68.5, which was obtained from the angular sweep SPR curve measured in water (Fig. 4b). This angle is the point with the highest SPR sensitivity to the changes in reflectivity due to binding events over the surface, (angle located in the middle of the maximum steepest slope of the SPR curve in water). Please refer to lines 300-305

The fig 4b shows the spr curve in water of a gold-coated chip before and after biofunctionalization, showing a shift, of 1.3 degrees, in the resonance angle due to the quantities of bound antibodies. Please refer to lines 315-327

6)     The authors must show more data to support their results. For example, they detailed some immobilization factors they studied (i.e.  SAM ratios and concentration of antibody). They just state that they select the conditions as they gave “maximum responses”. They must summarize the results for the other conditions as well, to support this conclusion and show some graphs/tables too. Looking at figure 3a. they do not indicate to what conditions (SAM and antibody concentration) this sensorgram corresponds to, but assuming is for the best ones, the immobilization signal is quite small. Moreover, if we check the detection in figure 4a. The concentrations used are considerably high and the signals very low, with a lot of noise.

The results of the immobilization factors tested during the study are summarize in an added graph (Figure 3).  According to the results obtained in Figure 3, the maximum response obtained for the immobilization was using a ratio 1:20 of MHDA: MUD and 50 µg mL-1 of anti-HF1. 

7)     The calibration curve in Figure 4 should be bigger and all the fitting parameters should be included. It also shows the detection of the protein at different concentration. The authors must clarify if each measurement has been obtained with a different chip or with the same. If it is the same chip, have they regenerated the surface? how? What conditions? And how stable is then the surface?

The calibration curve was improved and the R2 and the equation of the lineal regression was added. The measurements were performed employed the same chip. A regeneration solution of NaOH 20 mM was employed to remove the bound of the protein, obtaining a reusability of the surface for 10 cycles. This information has been incorporated to the manuscript, please refer to lines 268-270

8)     The matrix effect study is somehow incomplete. They must clarify if the extraction process they follow is the one that is commonly done to isolate the fungus, otherwise is useless. They do this process and afterwards, they spike the extract with the protein. In this case, the recovery is close to 100%. But it is not clearly explained if this process is enough to extract the fungus and what is the actual recovery of the methodology. Moreover, it is not clear if with this simple process is enough to detect the protein. Is it necessary to later extract also the protein from the cell wall? or are the antibodies capable of recognizing the protein directly in the fungus? Have this been evaluated? The authors should explain better this before assessing if this preliminary evaluation is useful or not.

The extraction process performed in this study is a methodology widely reported in previous works where the mechanical disintegration of the leaves allows the release of the fungus M. fijiensis, that normally colonizes the leaf tissue, growing intercellularly [7]–[9]. Furthermore, the antibodies are capable to detect directly the protein in the fungus, since the protein is exposed out, in the cell wall (unlike intracellular proteins). Please refer to lines 364-367

9)     The last paragraph before the conclusions they claim the potential of the method to achieve “early detection of the disease and that not sample pretreatment is needed”. This must be clarified in both terms: what are approximately the LOD requirements for early dtection? And what do they mean that no pretreatment is needed?

Early detection refers to the quantification of fungal proteins in leaves at a presymptomatic stage, discriminating from healthy leaves, in order to optimize fungicide applications for disease management, since the efficacy of the fungicide is improved in the earlier stages of the infection [9], [10]. However, the lack of studies addressing the evaluation of the amount of the inoculums in the leaves (the extension of primary infection in absence of symptoms) make difficult the establishment of initial concentrations, since the concentration depends on several factors such as the amount of leaf tissue selected, plant ageing, climatic conditions and influence of disease stage. Furthermore, Otero et al. [7] reported a monoclonal antibody-based on a triple antibody system-ELISA for the detection of Mycosphaerella fijiensis, obtaining a limit of detection of 10 µg mL-1, which is comparable with the one obtained in this study, 11.7 µg mL-1. On the other hand, in the study performed by Otero et al. asymptomatic leaves inoculated with the fungus were measured, finding concentration between 10 and 40 µg mL-1 of antigen. These results suggest the high potential of the analytical features in the SPR protocol proposed in this study for black sigatoka management [7]. Please refer to lines 341-343 and 374

In terms of the pretreatment of the sample, once the leaves extracts have been obtained, no further treatment of the sample is needed, such as dilution, centrifugation, purification or extraction steps. In contrast with other reported methods, such as ELISA and PCR.

Once again, I would like to thank for all the time and considerations you provide us in the reviewing of our work. I hope that our analysis and actions taken on your feedback are appropriate to clear up any of the doubts and comments stated by the reviewers.

Round  2

Reviewer 1 Report

Authors succeed to improve the quality of this work, but still there are many ponts that they decided to leave unrevised and they provide the arguments for that. The revision could be improved at least by inclusion of these arguments into the text of the manuscript. E.g. the explanation why they omit the XPS characterization of the layer and to (at least) provide the COOH density measured by other scientists, or the expected range of COOH densities according to the literature. Same arguments must be provided for antibody immobilization, i.e. more details explaining that from the SPR data it is eveident that ntibodies are immobilized and they are in active "form"...... Indeed, these observations were described before for other antibodies.

Finally, the remarks regarding the selectivity and sensitivity must be written in a clear way.

If authors have no results regarding the selectivity of this sensor, they should emphasize that this is ongoing work and provide some outlook in cconclusion section.

The manuscript still can be improved.

Author Response

Authors succeed to improve the quality of this work, but still there are many points that they decided to leave unrevised and they provide the arguments for that. The revision could be improved at least by inclusion of these arguments into the text of the manuscript. 

E.g. the explanation why they omit the XPS characterization of the layer and to (at least) provide the COOH density measured by other scientists, or the expected range of COOH densities according to the literature.

The reviewer has right since XPS is a traditional technique for surface characterization used to obtain chemical information about modified surfaces, however our research group does not have access to an XPS equipment. According to Cheng et al [1] the density of a mixed self-assembled monolayer (SAM) of alkanethiols (nitrilotriacetic acid thiol and oligo(ethylene glycol thiols), measured by XPS, ranged from 0.9 to 1.3 molecule/nm2 . In our work we solely evaluated the formation of the SAM with terminal COOH through an indirect methodology published by [2], [3], where the efficiency in the process of immobilization (which occurs through the activation of COOH moieties from the SAM) is measured in real time at fixed angle (represented in Fig 4a). In this case, the increment in the SPR reflectance corresponds to the increase of thickness in the film due to the density of immobilized molecules. The previous information was added, please refer to lines 284-291

Same arguments must be provided for antibody immobilization, i.e. more details explaining that from the SPR data it is evident that antibodies are immobilized and they are in active "form"...... Indeed, these observations were described before for other antibodies.

The entire process of immobilization of the antibodies was measured in real time as observed in (Fig 4a), where the increment in the SPR reflectance, at each step of the process, corresponds to the increase of thickness in the film due to the density of immobilized molecules [2], [3]. Furthermore, the angle of SPR curve, before and after the immobilization was measured, obtaining a shift of 1.3° after the immobilization of the antibodies (Fig 4b). According several studies, a shift of 0.1° in the response angle of a SPR curve corresponds to 1 ng/mm2 of biomolecule/protein immobilized [4]–[6], which in our work represents a density of 13 ng/mm2. Please refer to lines 324-30. Furthermore, the activity of the antibodies was measured directly through the immunoreaction antigen-antibody during the recognition assay of the protein from P. fijiensis (Fig 5), showing the bioactivity of the antibodies.

Finally, the remarks regarding the selectivity and sensitivity must be written in a clear way. If authors have no results regarding the selectivity of this sensor, they should emphasize that this is ongoing work and provide some outlook in conclusion section.

The sensitivity of the method was added to the manuscript and within table 1 (with the analytical parameters). In terms of the selectivity and the perspectives of future work were added to conclusion section. Please refer line 399-404

[1]      F. Cheng, L. J. Gamble, and D. G. Castner, “XPS, TOF-SIMS, NEXAFS, and SPR characterization of nitrilotriacetic acid-terminated self-assembled monolayers for controllable immobilization of proteins,” Anal. Chem., vol. 80, no. 7, pp. 2564–2573, 2008.

[2]      M.-C. Estevez, J. Belenguer, S. Gomez-Montes, J. Miralles, A. M. Escuela, A. Montoya, and L. M. Lechuga, “Indirect competitive immunoassay for the detection of fungicide Thiabendazole in whole orange samples by Surface Plasmon Resonance.,” Analyst, vol. 137, no. 23, pp. 5659–5665, Dec. 2012.

[3]      M. Soler, M.-C. Estevez, M. Alvarez, M. A. Otte, B. Sepulveda, and L. M. Lechuga, “Direct detection of protein biomarkers in human fluids using site-specific antibody immobilization strategies.,” Sensors, vol. 14, no. 2, pp. 2239–2258, Jan. 2014.

[4]      C. Y. Yang, E. Brooks, Y. Li, P. Denny, C. M. Ho, F. Qi, W. Shi, L. Wolinsky, B. Wu, D. T. Wong, and C. D. Montemagno, “Detection of picomolar levels of interleukin-8 in human saliva by SPR,” Lab Chip, vol. 5, no. 10, pp. 1017–1023, 2005.

[5]      H. Y. Song, X. Zhou, J. Hobley, and X. Su, “Comparative study of random and oriented antibody immobilization as measured by dual polarization interferometry and surface plasmon resonance spectroscopy,” Langmuir, vol. 28, no. 1, pp. 997–1004, 2011.

[6]      S. K. Vashist, C. K. Dixit, B. D. MacCraith, and R. O’Kennedy, “Effect of antibody immobilization strategies on the analytical performance of a surface plasmon resonance-based immunoassay,” Analyst, vol. 136, no. 21, pp. 4431–4436, 2011.
